# Regulatory Interplay between miR-181a-5p and Estrogen Receptor Signaling Cascade in Breast Cancer

**DOI:** 10.3390/cancers13030543

**Published:** 2021-02-01

**Authors:** Rosaria Benedetti, Chiara Papulino, Giulia Sgueglia, Ugo Chianese, Tommaso De Marchi, Francesco Iovino, Dante Rotili, Antonello Mai, Emma Niméus, Carmela Dell’ Aversana, Lucia Altucci

**Affiliations:** 1Department of Precision Medicine, University of Campania Luigi Vanvitelli, 80138 Naples, Italy; chiara.papulino@unicampania.it (C.P.); giulia.sgueglia@unicampania.it (G.S.); ugo.chianese@unicampania.it (U.C.); carmela.dellaversana@cnr.it (C.D.A.); 2Department of Oncology and Pathology, Lund University, SE-221 00 Lund, Sweden; tommaso.de_marchi@med.lu.se (T.D.M.); emma.nimeus@med.lu.se (E.N.); 3Department of Translational Medical Sciences, University of Campania “Luigi Vanvitelli”, Via L. De Crecchio 7, 80138 Naples, Italy; francesco.iovino@unicampania.it; 4Department of Drug Chemistry and Technologies, University of Roma ‘La Sapienza’, P.le A. Moro n. 5, 00185 Roma, Italy; dante.rotili@uniroma1.it (D.R.); antonello.mai@uniroma1.it (A.M.); 5Department of Surgery, Skånes University Hospital, 222 29 Lund, Sweden; 6Institute of Experimental Endocrinology and Oncology “Gaetano Salvatore” (IEOS)-National Research Council (CNR), 80131 Napoli, Italy

**Keywords:** miR-181a-5p, ERα, breast cancer, hormone signaling, epigenetic SERD, endocrine therapy

## Abstract

**Simple Summary:**

Despite huge efforts in breast cancer care programs, patient’s survival rates greatly vary. Differences in response to therapy still represent the major challenge for clinicians and biologists. Define new anticancer mechanisms and innovative predictors for resistance could be a valid strategy to permanently defeat breast cancer. Here we propose the epigenetic based reprogramming of breast cancer, which leverages on the crosstalk between miR-181a-5p and Estrogen Receptor α. This simultaneously approach allows to induce miR-181a-5p and reduce the receptor expression, blocking the estrogen-dependent proliferative pathway underlying breast cancer progression. Since the epigenetic approach insists on transcriptional regulation, it is mostly independent of the acquired resistance mechanisms typically induced by prolonged endocrine therapy and therefore can be used as a sensitizer, neoadjuvant, or in combination with the standard in care treatments against breast cancer.

**Abstract:**

The efficacy and side effects of endocrine therapy in breast cancer (BC) depend largely on estrogen receptor alpha (ERα) expression, the specific drug administered, and treatment scheduling. Although the benefits of endocrine therapy outweigh any adverse effects in the initial stages of BC, later- or advanced-stage tumors acquire resistance to treatments. The mechanisms underlying tumor resistance to therapy are still not well understood, posing a major challenge for BC patient care. Epigenetic regulation and miRNA expression may be involved in the switch from a treatment-sensitive to a treatment-resistant state and could provide a valid therapeutic strategy for ERα negative BC. Here, a hybrid lysine-specific histone demethylase inhibitor, MC3324, displaying selective estrogen receptor down-regulator-like activities in BC, was used to highlight the interplay between epigenetic and ERα signaling. MC3324 anticancer action is mediated by microRNA (miRNA) expression regulation, indicating an innovative function for this molecule. Integrated analysis suggests a crosstalk between estrogen signaling, ERα interactors, miRNAs, and their putative targets. Specifically, miR-181a-5p expression is regulated by MC3324 and has an impact on cellular levels of ERα. A comparison of breast tumor versus healthy mammary tissues confirmed the important role of miR-181a-5p in ERα regulation and points to its putative predictive function in BC therapy.

## 1. Introduction

Based on the 2018 GLOBOCAN report, the incidence of cancer is growing, and in the next years one out of five men and one out of six women will receive a cancer diagnosis, and one out of eight men and one out of ten women will die from the disease [1]. Breast cancer (BC) is the most commonly occurring cancer in women and the second most common cancer overall. There were over two million new cases of BC in 2018 [2] and according to incidence forecasts, BC will substantially increase over the next ten years [3]. Despite ongoing efforts to improve detection and treatment of BC for women worldwide, affected patients continue to experience an alarmingly high mortality rate, particularly in tumors exhibiting “ab initio” or acquired resistance to treatments [4]. Studies confirm that more than 70% of all BCs are estrogen receptor (ER)α-positive, meaning that cancer cells grow in response to the estrogen hormone [5]. These cancer types are typically treated with drugs such as tamoxifen (known as endocrine therapy or ER-targeted therapy) and aromatase inhibitors (AI), that either lower levels of the hormone or inhibit ERs in order to prevent the tumor from spreading [6]. However, around 35% of patients treated with endocrine therapy develop resistance, which negatively impacts overall survival [7]. The mechanisms underlying tumor resistance to therapy are not well understood and treatment resistance currently poses a major challenge. Epigenetic regulation and microRNA (miRNA) rebalance have been shown to be two key factors responsible for endocrine therapy resistance [8,9,10]. miRNAs regulate several cellular and signaling pathways, ranging from development and differentiation to cell proliferation and apoptosis [11]. In BC, dysregulation of a single miRNA or a small subset can therefore significantly impact cellular outcomes, possibly leading to the development of refractory forms of tumors [12]. miRNA profiling studies have identified deregulated miRNAs and their correlation with functions and molecular BC subtypes [13,14,15]. As miRNAs exert their effects at the translational level, they constitute an important link between coding genes and various cellular processes, taking part in the regulation of ~30% of all proteins [16]. In BC, this large regulatory mechanism is orchestrated by Erα, which regulates the expression of several miRNAs, potentially contributing to sustain the proliferation and pathological features of cancer cells, providing a specific signature for ERα-positive and ERα-negative BCs [17,18,19]. For this, miRNAs are promising biomarkers for endocrine therapy resistance prediction and patient stratification. On top of this, miRNA expression could be epigenetically regulated to block or reduce BC progression [20], providing an additional tool for therapeutic intervention. In this study, we describe an innovative mechanism that simultaneously modulates miRNA expression and ERα pathological function in BC cell lines and ex vivo patient samples. The gatekeeper mechanism here is modulation of the histone code, obtained using the lysine-specific histone demethylase (KDM) inhibitor MC3324, recently found by our group [21] to act as an “epigenetic” selective estrogen receptor down-regulator (SERD), showing anticancer activity in both in vitro and in vivo BC models [22].Via inhibition of LSD1 and UTX, MC3324 induces an epigenetic rebalance, which in turn has a major effect on mRNA, miRNA, and proteins. The extensive alterations in miRNA levels induced by MC3324 treatment play a key role in ERα functions and highlight the importance of miRNA expression in anti-estrogen resistance in BC. Our results show that interfering with regulation of the epigenetic code by blocking LSD1 and/or UTX has an immediate impact on the molecular fingerprint of BC, leading to down-regulation of estrogen-mediated pathways and cell death. Here, we describe the role of miRNA181-a-5p as a link between histone methylation changes and ERα-mediated survival pathways in BC.

## 2. Results

### 2.1. Inhibition of LSD1 and UTX Affects miRNA Profile

We previously demonstrated that MC3324 is a dual LSD1 and UTX inhibitor that negatively regulates ERα signaling and promotes activation of programmed cell death pathways in BC both sensitive and resistant to endocrine therapies [21]. Through the epigenetic and transcriptional reprogramming of BC cells, MC3324 acts as a SERD-like molecule. While carrying out a detailed investigation of the molecular mechanisms underlying BC efficacy, we found that one of the epigenetic modifications induced by MC3324 is associated with the expression of miRNAs. Based on this observation, we hypothesized that the miRNA regulatory network is dysregulated in BC hormonal signaling and might be rebalanced via epigenetic interventions. Comprehensive miRNA expression profiling was performed in the ERα-positive BC MCF-7 cell line, following the treatment with MC3324. miRNome profiling identified a subset of miRNAs modulated by LSD1 and UTX inhibition in BC (16 ≤ cycle threshold (CT) ≤ 37; 2^−ΔΔCT^ ≥ ±2). A total of 448 commonly expressed miRNAs were found in treated and untreated MCF-7 cells, as shown in Figure 1A. Of these, 96 miRNAs were only expressed in the control condition, as shown in Appendix A, while 26 miRNAs were enriched after treatment with MC3324, as shown in Appendix A. We identified a cluster of 185 differentially expressed miRNAs, as shown in Figure 1B. Among these common miRNAs, 153 were down-regulated and 32 were up-regulated after MC3324 treatment, as shown in Appendix A. Predicted common target genes of the 448 differentially regulated miRNAs were identified using the miRSystem database, which integrates seven well-known miRNA target gene prediction tools: DIANA [23], miRanda [24], miRBridge [25], PicTar [26], PITA [27], rna22 [28], and TargetScan [29]. Furthermore, all predicted targets were matched with genes differentially regulated by MC3324 identified using the whole transcriptome (RNAseq: GSE130067), as shown in Figure 1B [21]. Specifically, down-regulated genes (1763/2933) were compared with the predicted targets of up-regulated miRNAs (1451/3640), and up-regulated genes (1173/2933) with predicted targets of down-regulated miRNAs (3195/3640). The results are summarized in Figure 1C,D; 200 predicted target genes were confirmed as down-regulated and 282 predicted target genes as up-regulated, as shown in Appendix A. Functional annotation enrichment of the target gene subset was performed using the Molecular Signatures Database (MSigDB) in Gene Set Enrichment Analysis (GSEA, San Diego, CA, USA) v2.2.2 software. Gene Ontology terms enrichment indicated the crucial role of miRNA patterns in the complex regulation of key biological processes, such as transcription, cell death, differentiation, and response to hormone signaling, including ERα pathways, as shown in Figure 1E.

### 2.2. Integrative Analysis of Multi-Omics Data: miRNome, mRNA Transcriptome, and Interactome

The dynamic interplay between estrogen signaling cascade and miRNAs was obtained by the integration of transcriptome, ERα and LSD1 proteomic interactome, and miRNA expression profiling data. Differentially expressed miRNAs and related target genes were compared with MS/MS results (PXD012781).The lists of LSD1 and ERα interactors under- and over-represented after MC3324 induction for 6 and 24 h obtained by immunoprecipitation and MS/MS analysis were compared with: (i) 200 commonly predicted and validated (RNAseq transcriptome analysis) down-regulated genes; (ii) 282 commonly predicted and validated (RNAseq transcriptome analysis) up-regulated genes. We observed a correlative trend between miRNAs, target genes, and protein interactors after MC3324 treatment, as shown in Table 1 and Figure 2A. “Bona fide” down-regulated ERα interactors were ENAH, STRBP, TBC1D9, USP32, PKP4, OSBPL8, and TBL1XR1, while up-regulated interactors were SCD, RSBN1, and ARHGAP17, as shown in Figure 2A. Down-regulated LSD1 interactors were ANKRD50, CCDC6, DCLK1, TANC2, TBC1D1, and USP32, while up-regulated LSD1 ones were MAP2K7, MYH10, NAP1L1, and TFRC, as shown in Figure 2B. Some of these ERα and LSD1 interactors are involved in different cellular processes such as protein assembly and modification (e.g., USP32, NAP1L1), vesicular transport and remodeling (e.g., ENAH, STRBP), cytoskeleton organization (e.g., CCD6, DCLK1, TANC2, MYH10) [21], and intracellular protein transport (e.g., TBC1D1, TBC1D9, ANKRD50) [30]. The relative expression of miRNA targeting validated ERα and LSD1 interactors is provided in Figure 2C. Some of these ERα and LSD1 interactors were also validated in an MCF-7 cell line by qPCR, as shown in Appendix A.

### 2.3. ERα-Mediated miRNA Signature

Interestingly, we picked out that TBC1D9, USP32, and TBC1D1 are shared ERα and LSD1 interactors and are negatively regulated by the same miRNAs (miR-181a-5p, miR-181c-5p, and let-7f-5p) following MC3324 treatment, as shown in Table 1. This finding prompted us to focus on these three specific miRNAs, despite the huge effect on miRNome, as shown in Figure 1A–E. Moreover, a substantial number of differentially regulated miRNAs (13 down, 4 up) were associated with ERα signaling (direct targeting), and activation cascade, as shown in Figure 3A. Subsequently, we matched the data from Table 1 with miRNAs reported in Figure 3A, and we further restricted the number of miRNAs to be investigated to solely miR-181a-5p. The role of miR-181a-5p in BC is contradictory [31]. Several studies report that miR-181a-5p exerts anti-BC action, preventing tumor invasion and metastasis, reducing mammosphere formation, promoting cancer cell death, and favoring drug sensitivity [32,33,34], suggesting its oncogenic function. Other studies describe a down-regulation of miR-181a-5p in more aggressive or late-stage BC, indicating its role as a tumor suppressor gene [35]. To provide comprehensive evidence that LSD1 and UTX inhibition by MC3324 modulates the ERα signaling pathway and its interactome via functional activity of miR-181a-5p, we compared data in ERα-positive (MCF-7) and ERα-negative (MDA-MB-231) BC models, as shown in Figure 3B,C. Although expression levels of TBC1D1, TBC1D9, and USP32 genes decreased in MDA-MB-231 cells, as in MCF-7 cells, the confidence interval in the data is lower, as shown in Figure 3B. In MDA-MB-231, it is inconceivable to disregard the compensatory role played by ERβ, which is reported to be expressed independently of ERα [14,36]. Since ERα is a predicted target of miR-181a-5p, we investigated whether miR-181a-5p expression could be dependent on ERα expression or vice versa. ERα was absent in MDA-MB-231 cells and decreased in MCF-7 cells during MC3324 treatment in a time-dependent manner, as shown in Figure 3B. The basal expression level of miR-181a-5p quantified by qPCR was higher in triple negative BC (TNBC) MDA-MB-231 cells than in MCF-7 cells, as shown in Figure 4A. Following MC3324 induction, miR-181a-5p was up-regulated in MCF-7 cells while no significant differences were observed in MDA-MB-231 cells, as shown in Figure 4B,C. These findings suggest an important regulatory role for miR-181a-5p in the ERα pathway. The functional dependence between ERα and miR-181a-5p was further highlighted by comparison with BC models resistant and sensitive to endocrine therapy, as shown in Appendix A. In MCF7/TamR1, the ERα-miR-181a-5p pair is regulated in the opposite way to the MCF7 parental line, as well as the behavior of the ERα-regulated genes (e.g., PS2/TFF1). To assess whether ERα and its interactors can be regulated by miR-181a-5p, a miR-181a-5p mimic and inhibitor were transiently overexpressed in HeLa cells. qPCR analysis revealed an 8-fold increase in miR-181a-5p expression and ~70% of its down-regulation using the miRNA inhibitor compared to both miR-scramble and carrier controls, as shown in Figure 5A. miR-1 and let-7c were used as positive controls for miRNA mimic and inhibition approaches, respectively. qPCR analysis detected a 7-fold increase in mimic miR-1 expression and a 60% reduction in let-7c levels, as shown in Figure 5A. For both controls, we evaluated the expression of their known target genes, HMGA2 for let-7c and PTK9 for miR-1, as shown in Appendix A, confirming HMGA2 up-regulation via inhibition of let-7c and PTK9 down-regulation upon miR-1 overexpression. At 48 h after transfection, the expression of ERα and its well-known regulated gene PS2/TFF1 were strongly reduced upon overexpression of miR-181a-5p, as shown in Figure 5B and Appendix A. The inverse correlation between miR-181a-5p and ERα was further corroborated by up-regulation of target genes after inhibitor action, as shown in Figure 5B. We also evaluated the expression of USP32, TBC1D1, and TBC1D9 upon transfection of the miR-181a-5p mimic and inhibitor. TBC1D1, USP32, and TBC1D9 were down-regulated upon miR-181a-5p overexpression compared to miR-scramble and carrier controls, as shown in Figure 5B. Conversely, their expression levels significantly increased following miR-181a-5p inhibition, as shown in Figure 5B. These findings suggest that these genes are targets of miR-181a-5p. Both ERα and TBC1D9 expression levels were correlated and pathway analysis software provided additional evidence of gene/protein interactions [37]. TBC1D9 overexpression was also found increased in carcinomas of males compared to those of females and may therefore represent a novel molecular target for development of gender-specific therapeutics and companion diagnostics [38]. Out of gene regulatory networks analysis, TBC1D1 is part of special nodes in the basal B BC subtype together with miR-181d, one of four highly conserved mature family members (miR-181a, miR-181b, miR-181c, miR-181d) [31,39]. USP32 was also found overexpressed in BC cell lines and primary breast tumors [40], suggesting its role as a therapeutic and prognostic target in ERα-positive BC [41]. Further, stable silencing of USP32 expression reduced proliferation and migration in the ERα-positive MCF-7 cell line [40]. Our experimental model demonstrates USP32, TBC1D9, TBC1D1, and ERα down-regulation and the impairment of their interactors in the context of BC.

### 2.4. miR-181a-5p as a Potential Hallmark of BC

To validate the correlation between ERα and miR-181a-5p, 18ex vivo BC samples and their paired healthy tissues were analyzed. miR-181a-5p baseline expression profile was evaluated by qPCR, comparing tumor (*n* = 18 samples) vs. normal adjacent (*n* = 18 samples) breast counterparts. The results revealed a strong and significant miR-181a-5p down-regulation in 15 BC samples compared with healthy counterparts, while miR-181a-5p was up-regulated in only three tumor samples, as shown in Figure 6A. To assess whether miR-181a-5p expression levels change upon ERα down-regulation, single-cell suspensions from BC and healthy tissues were freshly prepared and treated with MC3324 for 24 h. Following MC3324 induction, miR-181a-5p was weakly/mildly up-regulated in 9/18 samples of healthy breast tissues, was unchanged in 5/18 samples, and was down-regulated in 4/18 samples, as shown in Figure 6B. In contrast, qPCR analysis in primary BC tissues confirmed miR-181a-5p induction in almost all primary tumor samples, corroborating the data obtained in MCF-7 cells, as shown in Figure 1C. Together, these data suggest a prominent role for miR-181a-5p in BC after MC3324 treatment and validate miR-181a-5p as a possible hallmark for breast cancer via ERα signaling. We also investigated a possible correlation with clinical and biological parameters of 18 primary BC tissues, as shown in Table 2. Ki-67, ERα-, progesterone (PR)*-* positivity in nuclei, and HER2 status were measured by immunohistochemistry (IHC) analysis on BC patient-derived samples. The percentages of ERα*-*positive and Ki-67-positive cells in all patient samples are reported in Figure 7A,B, respectively, and representative images of IHC staining for both proteins in three different primary BC tissues are shown in Figure 7C. Dividing the cohort of patients into two clusters of ERα-positive patients (pt #1–3, 5, 6, 9–12, 15–18) and ERα-negative patients (TNBC and tumors with at least positivity for PR and/or HER2; pt #4, 7, 8, 13, 14) revealed that, following treatment with MC3324, the fold variation of miR-181a-5p was lower when ERα-dependent signaling was impeded, as shown in Figure 8. In BC tissues, we detected a 73% increase in response in ERα-positive compared to ERα-negative patients. In contrast, in the panel of healthy tissues expressing ERα at physiological levels, an increase in miR-181a-5p was also observed in counterparts of ERα-negative tumors. The differential ratio showed an increase of about 38%, indicating that an epigenetic aspect/memory may be able to influence the response of healthy cells. These observations are in line with the clinical significance of BC subtypes. Together, these findings suggest that epigenetic regulation of miR-181a-5p is part of a regulatory feedback loop in which ERα protein plays an essential role. miR-181a-5p can be efficiently modulated when ERα is present and active inside BC cells. In TNBC (e.g., MDA-MB-231) or in ERα-negative ex vivo tumors, the inhibition of KDMs is less effective at inducing overexpression of miR-181a-5p.

## 3. Discussion

Our results corroborate and strengthen the fact that the modulation of KDM enzymes, obtained via MC3324 inhibition, directly regulates ERα in BC, as also previously reported in [21]. ERα activity controls the synthesis, maturation, and steady-state levels of a large number of miRNAs in BC cells. Reducing the activity of KDMs has an immediate impact on ERα expression and down-regulation, which, in turn, has an effect on miRNome and hormone signaling in BC cells. Specifically, miR-181a-5p seems to be one of the key regulators of ERα and its interactors. This is underlined by the evidence that the expression level of miR-181a-5p is dependent on KDM and ERα activity axes in BC. miR-181a-5p is one of the members of the miR-181s family, together with miR-181b, c, and d [42]. Published data identified alteration of intracellular and circulating miR-181a expression in tissue and in serum of BC patients [43]. The down-regulation of miR-181a-5p is also seen in multidrug resistant forms of MCF7 cells and in aggressive or late-stage BC [34]. Thus, on one hand, miR-181a-5p might play as a tumor suppressor gene, reducing mammosphere formation, inducing cancer cell death, and enhancing drug sensitivity. On the other hand, oncogenic activities have been reported when miR-181a-5p is up-regulated in BC (e.g., high miR-181a levels were associated with poor survival rates after treatment) [44]. This contradictory role will certainly need deeper analyses to be further clarified. Identifying the intracellular crosstalk between ERα and miR-181a-5p (and its predicted and validated targets) could provide new interventional points to decipher the pleiotropic actions of ERα and its role as an onco-promoter in breast carcinogenesis and tumor progression. Moreover, the correlation between miR-181a-5p expression in cancer and healthy breast tissues, coupled with BC histological classification, might represent a path to determine the tumor suppressing or tumor promoting action of miR-181a-5p. As reported in Figure 1, Figure 2 and Figure 3, miR-181a appears to directly target multiple genes involved in BC. Investigation of these interactions is required to determine the miR-181a targets and networks to advance into therapeutic strategies. The direct comparison between tumor and healthy tissue of each breast cancer patient provided in our study is certainly a strong point to address further investigations. A future rise in the number of samples may indeed help strengthen the correlation between miR-181a, Erα, and the clinical outcome of BC patients. Likewise, it is reasonable to conceive that targets and expression levels of miR-181a-5p could be used as molecular markers for prognosis of primary BC, for prediction of disease responsiveness to endocrine treatment over time, and/or for stratification of patients who may benefit from a future epigenetic approach as adjuvant therapy.

## 4. Materials and Methods

### 4.1. Cell Culture and Treatment

MCF-7 and MDA-MB-231 and MCF7/TamR1 cells (ATCC, Milan, Italy) were grown in supplemented Dulbecco’s modified Eagle medium (DMEM; EuroClone, Milan, Italy). Primary cells isolated by BC tissue biopsy were cultured in DMEM/F12 + 10% inactivated fetal bovine serum (FBS) (Sigma-Aldrich, St Louis, MO, USA). Culture protocols are reported in [21]. MC3324, synthesized by Professor Mai’s group (“Sapienza” University of Rome), was dissolved in dimethyl sulfoxide (Sigma-Aldrich, St Louis, MO, USA) and used at a final concentration of 25 μM. Tamoxifen (Sigma-Aldrich, St Louis, MO, USA) was dissolved in ethanol and used at 1 μM for 24 and 48 h.

### 4.2. miRNome Profiling

miRNAs were profiled by q-PCR using a miRCURY LNA™ Universal RT microRNA PCR System (Qiagen, Milan, Italy) according to the manufacturer’s instructions. Real-time PCR reactions were carried out on a 7900HT thermocycler (Applied Biosystems, Foster City, CA, USA) using the thermal cycling parameters suggested by the manufacturer’s protocol. Raw Ct values were calculated using RQ manager software v1.2.1 (ABI, Waltham, MA, USA) with manual settings for threshold and baseline. All data were analyzed using a ΔRn threshold of 60 and baseline subtraction using cycles 1–16. miRNA profiles were determined using the −ΔΔCt method.

### 4.3. RNA Isolation and miRNA Expression Validation

Total miRNA-enriched RNA was purified and miRNA expression levels were measured by real-time PCR as previously reported [45]. Briefly, cells were centrifuged and resuspended in 1 mL of TRIzol reagent (Invitrogen, Monza and Brianza, Italy), vortexed, and stored overnight at −20 °C. Then, 100 µL of 2-bromo-3-chloro propane (Sigma Aldrich) were added to the samples, gently shaken, and incubated for 15 min at room temperature RT. After centrifugation at 12,000 rpm for 15 min at 4 °C, the supernatants were collected in fresh tubes supplemented with 500 µL of cold isopropyl alcohol. The total RNA was precipitated for 30 min at −80 °C followed by a centrifugation of 30 min at 12,000 rpm at 4 °C. The samples were resuspended in 1 mL of cold 70% ethanol and then centrifuged for 10 min at 7500 rpm at 4 °C. The RNA pellets were dried at 42 °C for a few minutes and resuspended in DEPC-treated H2O. The RNA samples were quantified using Nanodrop 1000 and their quality was checked using an Agilent RNA 6000 Nano Assay.

The miRNA fraction was converted into cDNA using miScript Reverse Transcription Kit (Qiagen), following the supplier’s instructions. miRNA Real-Time PCR was performed with QuantiTect SYBR Green PCR Kit (Qiagen) using 75 ng of cDNA in presence of 1× QuantiTect SYBR Green PCR Master Mix, miScript Universal Primer and primer specific for miR-181a-5p (Qiagen), let7-c (Qiagen), and miR-1 (Qiagen). RNU6b (Qiagen) specific primer was used to normalize data.

### 4.4. Gene Expression Analysis by qPCR Quantitative Real-Time PCR

Real-time RT-PCR was performed to examine mRNA expression levels using the VILO cDNA Synthesis Kit (Invitrogen, Monza and Brianza, Italy) to convert RNA into cDNA. A 1X SYBR Green PCR Master Mix (Bio-Rad, Segrate, Milan, Italy) was used according to the manufacturer’s instructions, using 50 ng of cDNA. Primers used are listed in Appendix A.

### 4.5. Pre-miR Precursor and Inhibitor of miR-181a-5p Reverse Transfection

Pre-miR precursor and inhibitor reverse transfection was performed in HeLa cells (4.6 × 10^5^ cells for each experimental point) with siPORT Amine Transfection Reagent (Ambion, Waltham, MA, USA), following the manufacturer’s instructions. Reverse transfection was performed with 100 nM of miR-181a-5p Pre-miR precursor or inhibitor; 100 nM mimic miR-scramble and 100 nM inhibitor miR-scramble were used as negative controls.

### 4.6. Computational Prediction and Gene Enrichment Analysis of miRNA Target Genes

Target gene prediction of differentially expressed miRNAs was performed using the miRSystem database. All miRNA entries are annotated according to the latest miRBase (release 22) (http://mirbase.org/). Inclusion criteria were: common target genes in at least four tools (including validated genes), HIT ≥ 4, and observed/expected ratio ≥2. Gene Ontology terms were analyzed using the MSigDBv6.2 in Gene Set Enrichment Analysis (GSEA) software (http://software.broadinstitute.org/gsea/msigdb) for Annotation, Visualization, and Integrated Discovery.

### 4.7. Data Collection and Sampling of Primary BC Tissue

Ex vivo breast samples were obtained from the University of Campania “Luigi Vanvitelli” Hospital Department of Surgery in collaboration with Dr. Iovino. The Ethics Committee of the University of Campania “Luigi Vanvitelli” Hospital ap-proved the study (Prot. number: 24713). Data collection and sampling was performed for 18 primary BC tissues. Ex vivo cells from BC biopsies before and after MC3324 treatment (25 μM, 24 h) were isolated according to the protocol reported in [21].

### 4.8. Western Blot Analysis

Treated and untreated cell pellets were suspended in lysis buffer (50 mmol/L Tris-HCl, pH 7.4, 150 mmol/L NaCl, 1% NP40, 10 mmol/L NaF, 1 mmol/L PMSF, and protease inhibitor cocktail). The lysis reaction was carried out for 15 min at 4 °C. The samples were then centrifuged at 13,000 rpm for 30 min at 4 °C and protein concentration quantified by Bradford assay (Bio-Rad). After centrifugation, 50 μg of each sample were loaded on 10% of polyacrylamide gels and electroblotting on nitrocellulose membrane. Immunoreactive signals were detected with a horseradish peroxidase-conjugated secondary antibody (Bio-Rad). All the antibodies were used according to the manufacturer’s protocol. Antibodies used were: ER (sc-543), USP32 (sc-374465), TBC1D1 (ab229504), TBC1D9 (Bethyl A301-027A), and GAPDH (sc-365062). Semi-quantitative analysis was performed using ImageJ software.

### 4.9. IHC Evaluation

IHC primary antibodies used were: ERα (clone SP1, Ventana), PR (clone 1E2, Ventana), Ki67 (Clone 30–9, Ventana), Her2 (clone 4B5, Ventana). Tumors were considered positive for ER and PR when at least 1% of tumor cells showed unequivocal nuclear staining according to American Society of Clinical Oncology/College of American Pathologists (ASCO/CAP) guidelines. PR expression was considered high in the presence of nuclear staining in 20% or more cells. We set a cut-off point to distinguish low versus high Ki67 expression at 20%. The original HER2/neu immunostained glass slides were concurrently reviewed by pathologists at a multiheaded microscope, and the consensus HER2/neu immunoreactivity was manually scored by conventional microscopy as 0, 1+, 2+, or 3+ according to the proposed HER/neu scoring system for breast cancer. According to the percentage of stained malignant cells, criteria for HER2/neu score assignment were: 0, no staining or staining in <10% of cells; 1, faint staining in ≥10% of cells; 2, moderate staining in ≥10% of cells; and 3, strong staining in ≥10% of cells. Tumors classified as 0, 1+, and 2+ were considered “negative” and those scored as 3+ were classified as “positive”.

## 5. Conclusions

In breast cancer, the function of miRNAs as targets or biomarkers of response to anticancer activity has long been investigated. An even more important role is played by ERα in breast cancer, whose expression, according to other ancillary parameters, determines the choice and the therapeutic regimen for patients. In this study, we highlight the existence of a functional correlation between miR-181a-5p and ERα. This balance can be epigenetically regulated, through the use of a modulator of the KDM enzyme class. The study, therefore, hypothesizes in the future to use epigenetic modulation as an additional level of therapeutic intervention or as neoadjuvant therapy, able to limit and/or overcome the resistance mechanisms that often accompany the prolonged use of endocrine modulators.

## Figures and Tables

**Figure 1 cancers-13-00543-f001:**
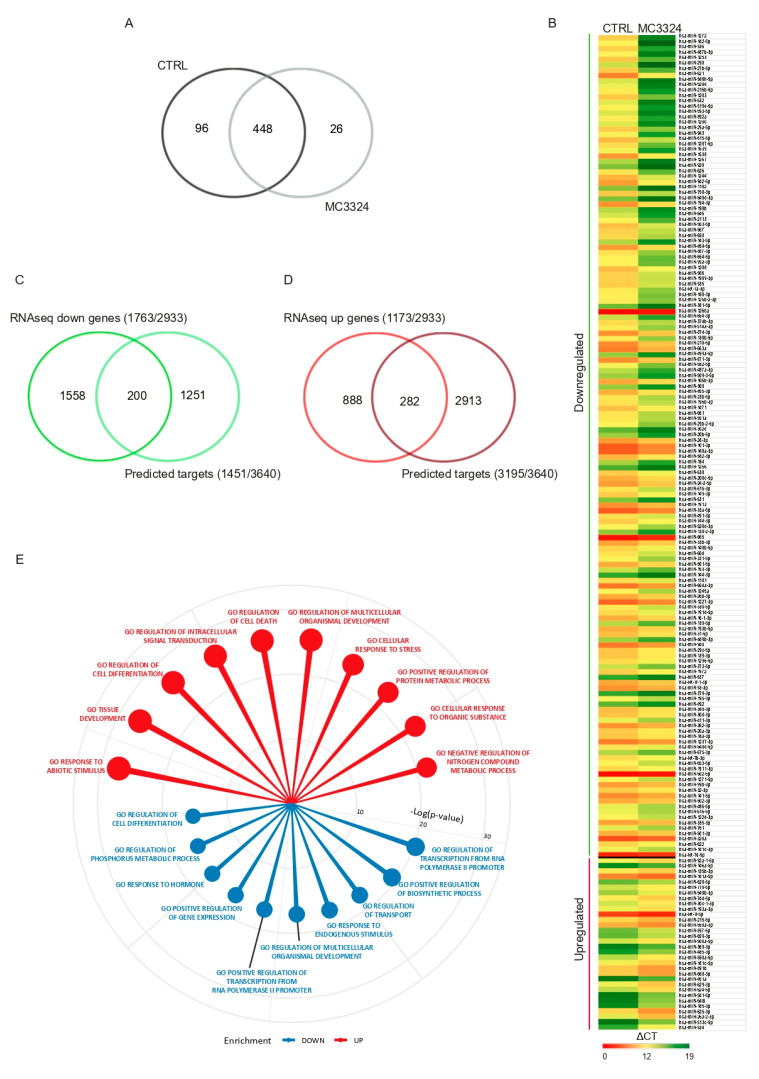
LSD1 and UTX inhibition regulates miRNome profile in breast cancer (BC). (**A**) Venn diagram and (**B**) heat map showing the 448 commonly expressed miRNAs in MC3324-treated and untreated MCF-7 cells. (**C**,**D**) Venn diagrams showing the predicted target genes of up- and down-regulated miRNAs validated by RNAseq analysis. (**E**) Gene Ontology enrichment analysis of up- and down-regulated target genes.

**Figure 2 cancers-13-00543-f002:**
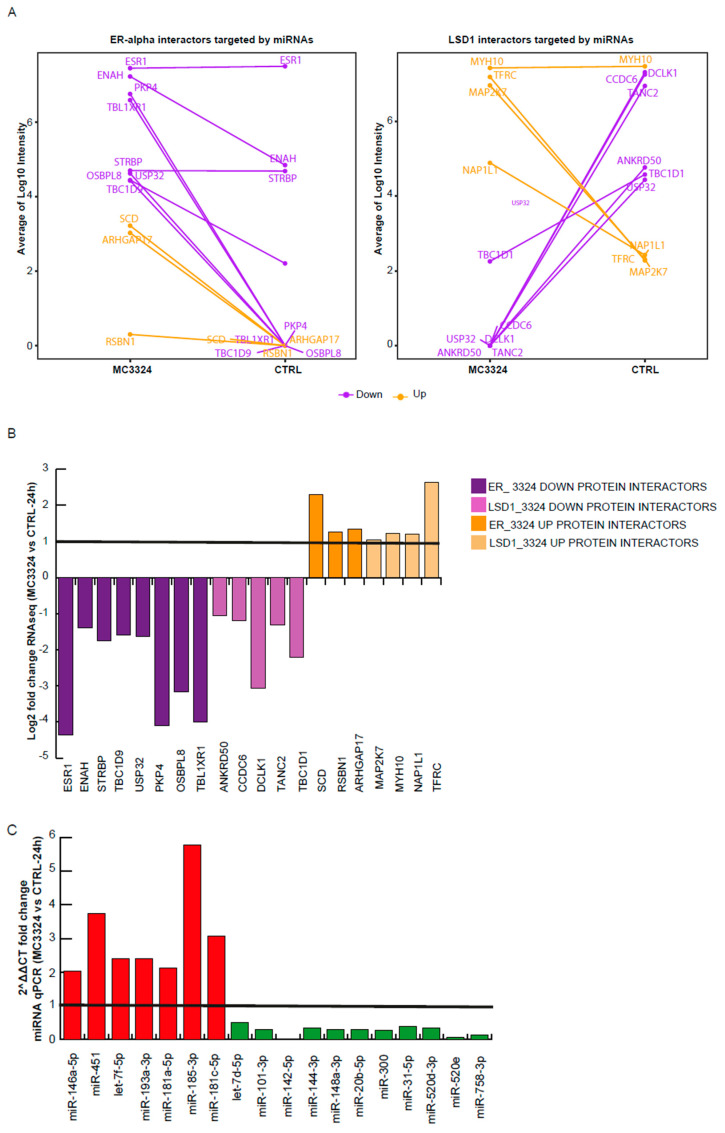
Integrative analysis of multi-omics data. (**A**) Average Log10 intensity of ERα and LSD1 interactors targeted by miRNAs. (**B**) Correlative trend between miRNAs, target genes, and protein interactors after MC3324 treatment. Log2 fold change (RNAseq) of MCF-7 cells untreated or treated with MC3324 for 24 h. (**C**) Relative expression levels of miRNAs targeting the validated ERα and LSD1 interactors.

**Figure 3 cancers-13-00543-f003:**
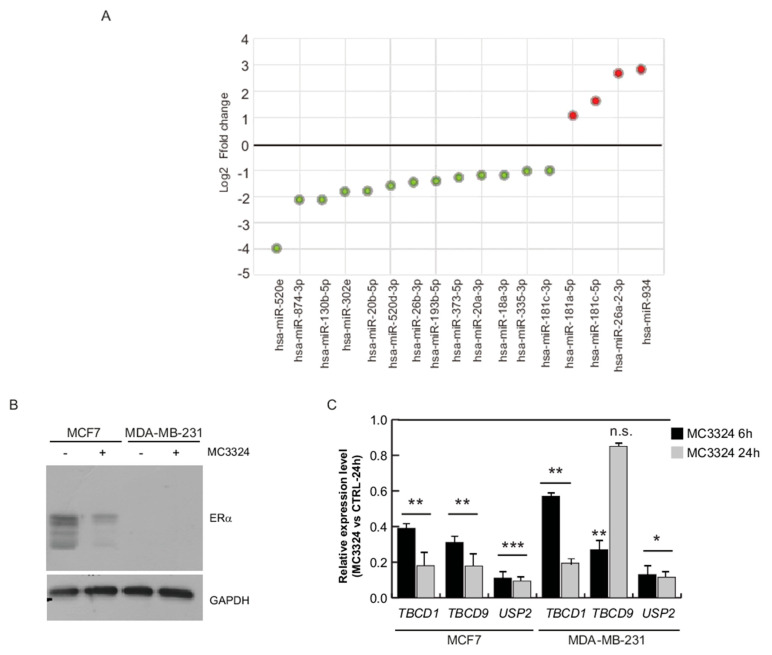
miRNA-mediated activity of MC3324 on ERα expression. (**A**) Log2fold change of expression levels of different BC-correlated miRNAs from literature and found modulated in MCF-7 cells upon MC3324 treatment (25 µM, 24 h). Green and red dots represent down-regulated and up-regulated miRNAs, respectively. (**B**) Western blot showing expression levels of ERα in MCF-7 and MDA-MB-231 cells before and upon MC3324 treatment (25 µM, 24 h). (**C**) Expression of USP32, TBC1D1, and TBC1D9 after MC3324 induction. MCF-7 and MDA-MB-231 cells were treated with MC3324 for 6 and 24 h at 25 µM. The results of three independent experiments each with three replicates are represented as the mean ± SEM (standard error of the mean).* *p* ≤ 0.05; ** *p* ≤ 0.01; *** *p* ≤ 0.001; n.s. for not significant variation.

**Figure 4 cancers-13-00543-f004:**
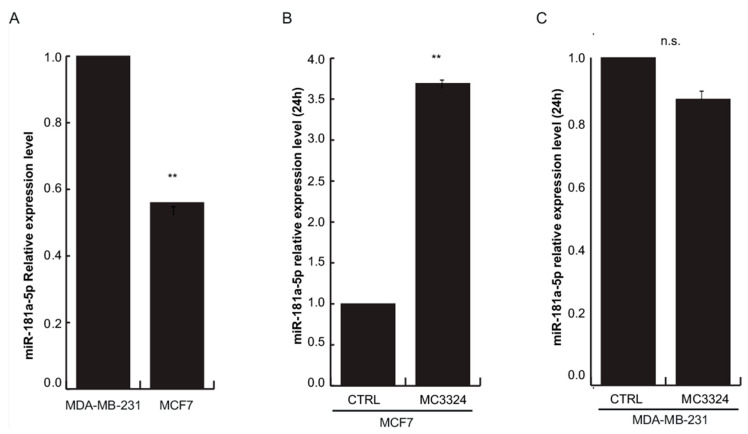
miR-181a-5p correlates with ERα expression. (**A**) miR-181a-5p relative expression levels in MCF-7 cells compared with triple negative BC (TNBC) MDA-MB-231 cells; (**B**) in MCF-7 and (**C**) in MDA-MB-231 cells upon MC3324 treatment (25 µM, 24 h). The results of three independent experiments each with three replicates are represented as the mean ± SEM. ** *p* ≤ 0.01; n.s. for not significant variation.

**Figure 5 cancers-13-00543-f005:**
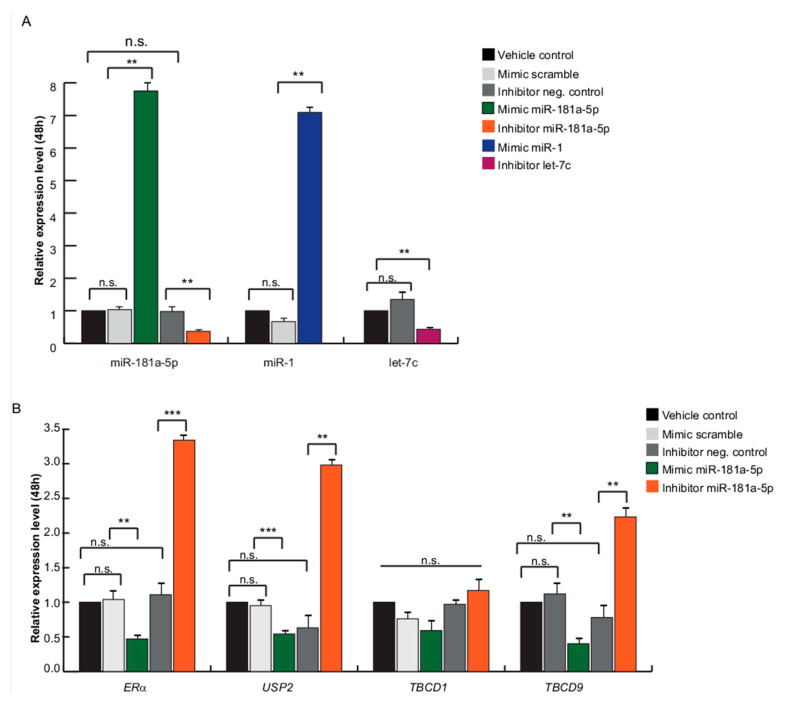
miR-181a-5p target validation. (**A**) miRNA relative expression levels determined by qPCR after transfection with synthetic mimic and inhibitor of miR-181a-5p and controls at a concentration of 100 nM for 48 h. (**B**) Relative expression levels of predicted target genes of miR-181a-5p determined by qPCR. RNU6B and GAPDH were used for data normalization for miRNA and gene expression. The results of three independent experiments each performed in triplicate are represented as the mean ± SD. ** *p* ≤ 0.01; *** *p* ≤ 0.001; n.s. for not significant variation

**Figure 6 cancers-13-00543-f006:**
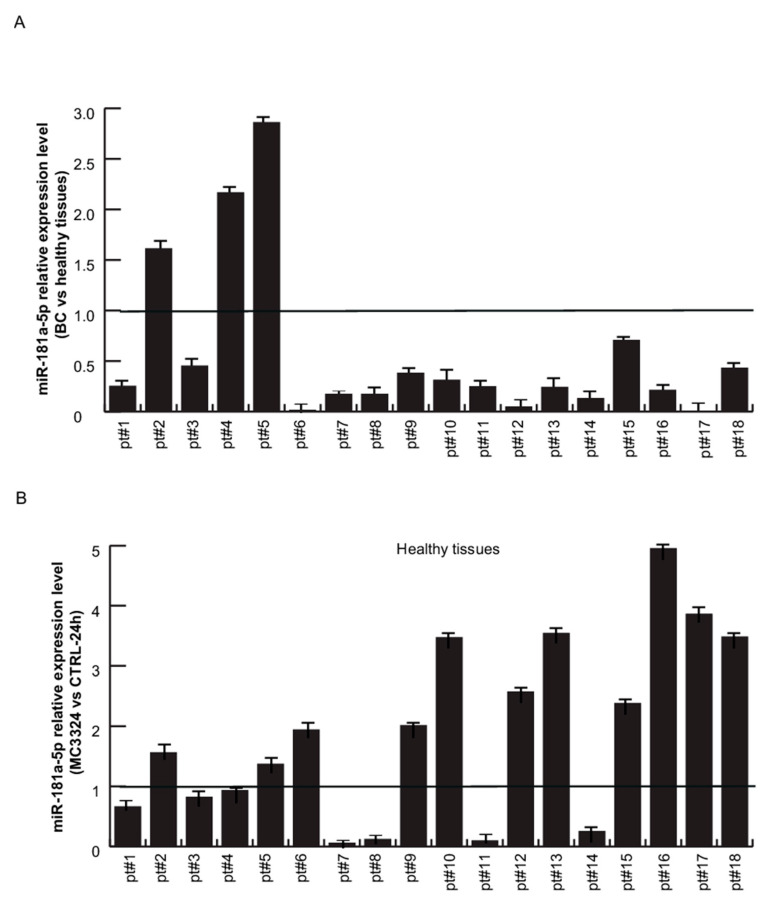
miR-181a-5p relative expression levels in 18 primary BC and healthy tissues.(**A**) miR-181a-5p relative expression levels in BC vs. healthy counterpart tissues. (**B**) miR-181a-5p relative expression levels after MC3324 treatment (25 µM, 24 h) in healthy breast and (**C**) BC tissues. The results are represented as the mean ± SD.

**Figure 7 cancers-13-00543-f007:**
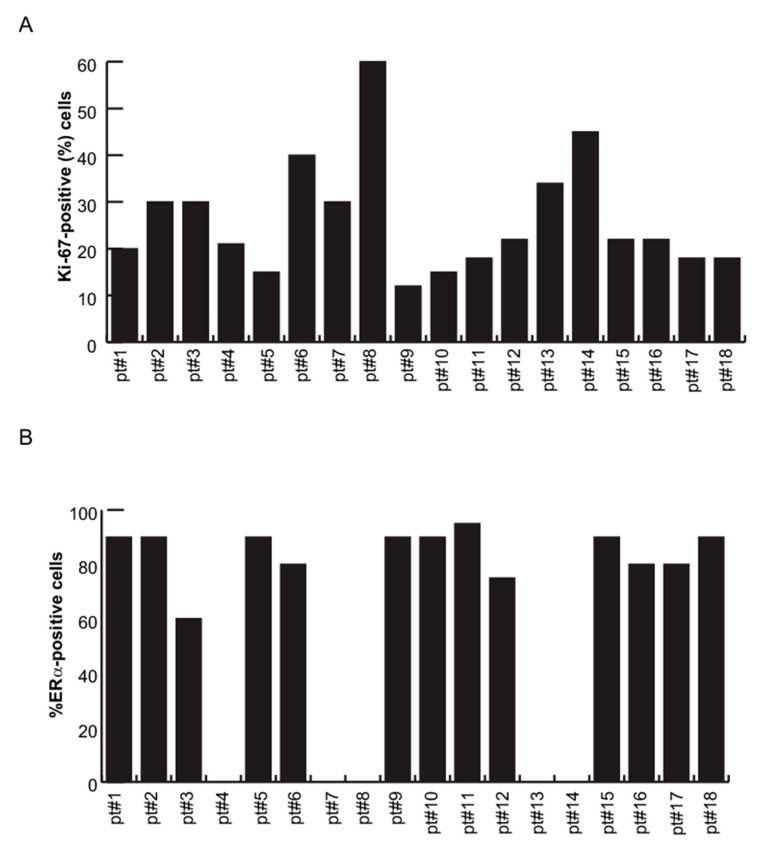
Immunohistochemistry of Ki-67 and ERα in 18 primary BC tissues. (**A**) Percentage of ERα*-*positive and (**B**) Ki-67-positive cells in 18 primary BC tissues. (**C**) Representative images of immunohistochemistry (IHC) staining for Ki-67 and ERα proteins in BC primary tissues (pt #7, 9, 12).

**Figure 8 cancers-13-00543-f008:**
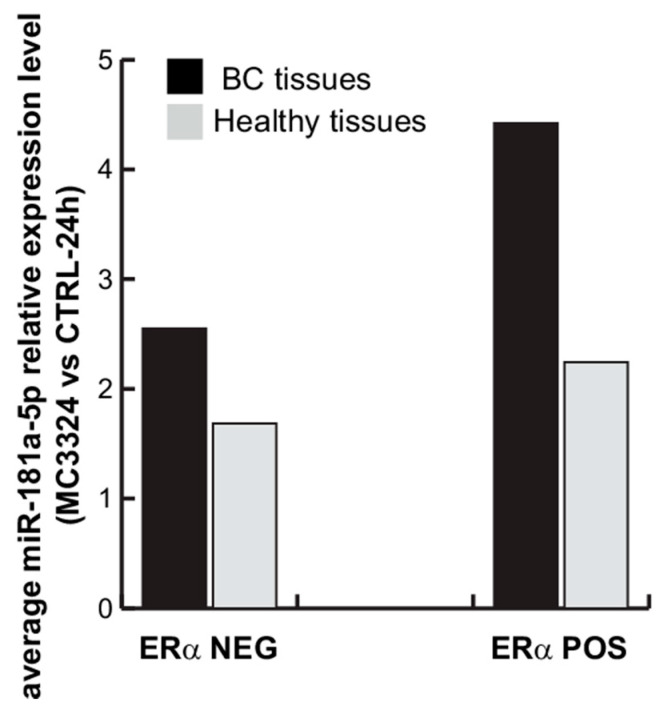
Average miR-181a-5p relative expression levels in 18 BC and healthy primary tissues (MC3324 vs. CTRL; 24 h). Grouping ERα-positive patients (pt #1–3, 5, 6, 9–12, 15–18) and ERα-negative patients (TNBC and tumors with at least positivity for progesterone (PR) and/or HER2; pt #4, 7, 8, 13, 14).

**Table 1 cancers-13-00543-t001:** Correlative trends between miRNAs, target genes, and ERα and LSD1 protein interactors in MCF-7 cells after MC3324 treatment.

	Gene Target	miRNA			
**ER_3324 DOWN PROTEIN INTERACTORS**	ESR1 (ERα)	miR-181c-5p	miR-185-3p		
	ENAH	miR-181a-5p	miR-181c-5p		
	STRBP	miR-181a-5p			
	TBC1D9	miR-181a-5p	miR-181c-5p	miR-451	
	USP32	let-7f-5p			
	PKP4	miR-193a-3p			
	OSBPL8	miR-181a-5p	miR-181c-5p		
	TBL1XR1	miR-181a-5p	miR-181c-5p	miR-193a-3p	
**LSD1_3324 DOWN PROTEIN INTERACTORS**	ANKRD50	miR-181a-5p	miR-181c-5p		
	CCDC6	miR-181a-5p	miR-181c-5p		
	DCLK1	miR-181a-5p	miR-181c-5p		
	TANC2	miR-146a-5p	miR-181a-5p	miR-181c-5	
	TBC1D1	miR-181a-5p	miR-181c-5p		
	USP32	let-7f-5p			
**ER_3324 UP PROTEIN INTERACTORS**	SCD	let-7d-5p			
	RSBN1	miR-20b-5p	miR-31-5p	miR-520d-3p	miR-520e
	ARHGAP17	miR-101-3p			
**LSD1_3324 UP PROTEIN INTERACTORS**	MAP2K7	miR-142-5p			
	MYH10	miR-300			
	NAP1L1	let-7d-5p	miR-101-3p	miR-148a-3p	
	TFRC	miR-144-3p	miR-148a-3p	miR-31-5p	miR-758-3p

**Table 2 cancers-13-00543-t002:** Clinical and biological features of 18 primary BC tissues.

	Cytological Classification	IHC Classification	Grade	Staging	PR (%)	HER2
PT#1	C5	Invasive carcinoma of no special type (NST)	G2	pT1N0	70	3+
PT#2		Invasive carcinoma of no special type (NST)	G2	pT1N0	70	0
PT#3		Invasive carcinoma of no special type (NST)	G3	pT2N2	60	1+
PT#4		Invasive carcinoma of no special type (NST)	G3	pT4bN0	0	3+
PT#5		Invasive carcinoma of no special type (NST)	G2	pT1N1a	0.8	1+
PT#6		Invasive carcinoma of no special type (NST)	G3	pT4bN2a	0.65	1+
PT#7		Invasive carcinoma of no special type (NST)	G2	pT2N1a	0	0
PT#8		Invasive apocrine carcinoma (IAC)	G3	pT2N1a	0	3+
PT#9		Invasive mucinous carcinoma (IMC)	G2	pT1cN0	0.7	0
PT#10		Invasive Ductal Carcinoma (IDC)	G1	pT2N0	0.03	0
PT#11		Invasive carcinoma of no special type (NST)	G2	pT2N1	0.9	1+
PT#12		Invasive Ductal Carcinoma (IDC)	G3	pT2N0	0.7	1+
PT#13		Invasive Ductal Carcinoma (IDC)	G2	pT1cN0	0	0
PT#14		Invasive Ductal Carcinoma (IDC)	G3	pT1bN1a	0	2+
PT#15		Invasive Ductal Carcinoma (IDC)	G2	pT2N2	<5	1+
PT#16		Invasive Ductal Carcinoma (IDC)	G2	pT1cN0	0.8	1+
PT#17		Invasive Ductal Carcinoma (IDC)	G2	pT1cN0	0.6	0
PT#18		Invasive Ductal Carcinoma (IDC)	G2	pT2N1M1 (bone)	0.7	0

## Data Availability

The data presented in this study are available on request from the corresponding authors.

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
