# Peer review of "Regulatory Interplay between miR-181a-5p and Estrogen Receptor Signaling Cascade in Breast Cancer"

_cancers, 2021, doi:10.3390/cancers13030543_

Round 1
Reviewer 1 Report
The manuscript entitled “Regulatory interplay between miR-181a-5p and estrogen receptor signaling cascade in breast cancer” is well written and presented. The authors show the potential of miR-181a-5p as a BC hallmark. The experiments are well designed although 18 samples are a low number for solid conclusions on ex vivo samples.
Line 282-283: please rephrase “might be playing” or “might play”
Line 317: how was miRNA isolated? Which method/kit?
Author Response
We would like to thank the reviewers for their thoughtful review of the manuscript. They raise important issues and their inputs are very helpful for improving the manuscript. We agree with almost all their comments and we have revised our manuscript accordingly. We are already crafting a revised version of the paper that states the hypothesis and the implications of our work more clearly than before. Moreover, we are including all reviewers’ suggestions and clarifying the text when needed. We are confident that the new version of the manuscript will be greatly improved. We respond below in detail to each of the reviewer’s comments and we include how we have revised things. We hope that the reviewers will find our responses to their comments satisfactory, and we are willing to finish the revised version of the manuscript including any further suggestions that the reviewers may have.
Point by point answers to Reviewer 1: Comments and Suggestions for Authors
We would like to thank the reviewer for the thoughtful comments and for the time spent reading this review. As the reviewer noted 18 samples certainly are a small number. The advantage in our case is to have tumors and healthy tissues from the same patient. Furthermore, in our work, we highlight how an epigenetic molecule can interfere simultaneously with the hormone pathway, transcription, and miRNA expression.
- Line 282-283: please rephrase “might be playing” or “might play”.
The correction was made.
- Line 317: how was miRNA isolated? Which method/kit?
Specification on materials and Methods were added.
Reviewer 2 Report
The manuscript by Benedetti et al describe how the expression of miR-181a-5p regulates cellular ERα expression, and further suggests miR-181a-5p as a molecular marker and a prognosticator for breast cancer endocrine resistance. However, the results described in the manuscript are not strong enough to support the conclusions above. To publish, a major revision should be recommended and the following experiments should be considered.
- Extra validation of how miR-181a-5p regulating ERα protein expression in cell lines is recommended, for example transfection of pri-miR constructor into cell lines and confirm again the expression level of ERα.
- It is not very clear whether ERα is the target for miR-181a-5p, therefore more experiments to investigate it is needed. For example, using reporter constructs and luciferase assays.
- The discussion suggests to use miR-181a-5p for endocrine treatment prediction, therefore a functional assay is desired to explore whether miR-181a-5p could sensitize ER positive cell line against tamoxifen treatment for example.
- Validate the results in another ER positive breast cancer cell line is highly recommended.
- In Fig8, p value is missing.
- Materials and methods section is not very clearly written, more information in details need to be provided. For example, the description of statistics, antibodies used to stain ER, PR, Ki67 and Her2.
Author Response
We would like to thank the reviewers for their thoughtful review of the manuscript. They raise important issues and their inputs are very helpful for improving the manuscript. We agree with almost all their comments and we have revised our manuscript accordingly. We are already crafting a revised version of the paper that states the hypothesis and the implications of our work more clearly than before. Moreover, we are including all reviewers’ suggestions and clarifying the text when needed. We are confident that the new version of the manuscript will be greatly improved. We respond below in detail to each of the reviewer’s comments and we include how we have revised things. We hope that the reviewers will find our responses to their comments satisfactory, and we are willing to finish the revised version of the manuscript including any further suggestions that the reviewers may have.
Point by point answers to Reviewer 2: Comments and Suggestions for Authors
We would like to thank the reviewer for his/her thoughtful comments and efforts towards improving our manuscript. In the following, we highlight the general concerns of the reviewer and our effort to address these concerns. We then address comments specific to each point below.
- Extra validation of how miR-181a-5p regulating ERα protein expression in cell lines is recommended, for example transfection of pri-miR constructor into cell lines and confirm again the expression level of ERα.
This experiment is present in the manuscript. As expected, the overexpression of miR-181a-5p is accompanied by the simulated down-regulation of ERα. The functional link between the two players is also confirmed by the down-regulation of the interactors (Figure 5).
- It is not very clear whether ERαis the target for miR-181a-5p, therefore more experiments to investigate it is needed. For example, using reporter constructs and luciferase assays.
Several predictor tools were used to identify the validated targets of hsa-miR-181a-5p. In particular, our results derive from miRSystem database which integrates seven well known miRNA target gene prediction programs: DIANA, miRanda, miRBridge, PicTar, PITA, rna22, and TargetScan. We annotated all miRNA entries according to the latest miRBase (release 22) (http://mirbase.org/) and used qualificatoryprediction target inclusion criteria (common target genes in at least four tools-including validated genes- HIT ≥>4, and observed/expected ratio ≥2). There are several shreds of evidence in the literature reporting the functional link between miR-181a-5p and ERα. The direct regulation of Erαby the miR-181a-5p has been reportedin myometrial system and mouse astrocytes (J ClinEndocrinolMetab. 2016 Oct;101(10):3646-3656; Mol Cell Neurosci. 2017 Jul;82:118-125). Moreover, miR181a (but also b and d) contains ERα binding sites in the regulatory region; it is transcriptionally repressed by estradiol-induced ER and activated following tamoxifen treatment (Cancer Res. 2009;69:8332–40; Proceedings of the National Academy of Sciences U S A. 2009;106:15732–7; Oncogene. 2015;34:2309–16). In the Oncogene article (Oncogene. 2009; 28: 3926–36), miR-181a-5p has the same behavior in both MCF7 and BT474, supporting the idea that ERα could be the target.In our paper we support these findings and, in addition, we propose a scenario in which the treatment with an epigenetic modulator (in this case a KDM inhibitor), can influence on the one hand the expression of ERα (Cancers (Basel). 2019 Dec 16;11(12):2027) and on the other hand the signature of miRNA. The identification of the ERα-miR-181a-5p pair and the simultaneous regulation of both allow increasing the efficacy of epigenetic based anti-cancer in breast cancer.
To clarify the reviewer’s doubt,we analysed the expression of a direct target gene of ERα: PS2/ TFF1.In Supplementary Figure 3, we further corroborate our hypothesis of direct regulation between miR-181a-5p and ERα analysing the PS2 / TFF1 expression level upon miR-181a-5p mimic- and inhibitor.
- The discussion suggests using miR-181a-5p for endocrine treatment prediction, therefore a functional assay is desired to explore whether miR-181a-5p could sensitize ER positive cell line against tamoxifen treatment for example.
We have taken reviewer’s comment in full consideration and it will be well reflected by the revised version of manuscript. For this reason we addeda new experiment reporting the different levels of expression of miR-181a-5p in two cellular models: MCF7 and MCF7 TamR1. MCF7 TamR1 is a tamoxifen-resistant cell line derived from the MCF-7/S0.5 human breast cancer cell line by long term treatment with the drug tamoxifen (ATCC CRL-3435TM). As it is shownin the new added Supplementary Figure 2 A-C, despite in both MCF7 and MCF7/TamR1 cells ERα expression increases following tamoxifen treatment (A), in MCF7/TamR1 PS2/TFF1 expression is unchanged (B). In the resistance to endocrine treatment condition miR-181a-5p is down regulated in a time dependent manner (C). It can be further noted that the expression trend of miR-181a-5p, following treatment with tamoxifen, is opposite between the MCF7 and MCF7/TamR1 cell lines.This observation implies and re-emphasizes the functional relationship between miR-181a-5p and ERα in breast cancer.
- Validate the results in another ER positive breast cancer cell line is highly recommended.
Accordingly to reviewer suggestion,we added the MCF7 TamR1 cell linewhich isanERα positive and tamoxifen resistant breast cancer model.
- In Fig8, p value is missing.
Figure 8 reports the difference in miR-181a-5p expression in ex-vivo samples. The figure suggests also that there is a certain degree of epigenetic modulation whether cancer samples are ERα+ or ERα-. As it can be observed from the information reported in Table 2, the ex vivo samples used in this study have different histological characteristics. Furthermore, the 18 samples are not equally distributed between ERα+ and ERα-. The inclusion of these ex vivo samples in our manuscript is still useful because, for each sample, we have a healthy tissue counterpart. The possibility of comparing the levels of miR-181a-5p and ERα between tumor and healthy tissues of the same patient and of epigenetically treating the cells deriving from the dissociation of both tissues, allows us to speculate on the existence of a functional, epigenetically drugable relationship between miR-181a-5p and ERα.
In any case, since we are aware that 18 is a small number, we did not feel like giving statistical significance to our data. However, we used the cohort we have available to confirm what it was found in the cell lines and to suggest a possible future development of the study.
- Materials and methods section is not very clearly written, more information in details need to be provided. For example, the description of statistics, antibodies used to stain ER, PR, Ki67 and Her2.
Material and Methods were updated with product specifications. The new experiments were also described accordingly. We upgraded Table 6 with primer sequences for PS2/TFF1:
PS2/TFF1 FORWARD sequence: GCTTAGGCCTAGACGGAATGGGC
PS2/TFF1 REVERSE sequence: CCAGGTCCTACTCATATCTGAGAG
Round 2
Reviewer 2 Report
Major comments:
The authors have responded sufficiently to some questions/concerns. Still, major concerns remain. MCF TAmR is a variant of MCF7wt and should not be considered a suitable ER+ second cell line for validation purpose. T47D or similar ER+ cell line would be more appropriate. I strongly suggest that the most important experiments are reproduced in at least one other cell line. See comment 4 above.
Table 2: The table is full of errors. Why showing PR but not ER? eg PT 10 with 0,03% !?! Furthermore, in methods, HER2 assessment is not correct. Please consult a pathologist.
Author Response
Reviewer 2 specific points: As indicated by Reviewer 2, theminor changes that remain to be addressed before the final acceptance of themanuscript are the revision of Table 2 (i.e. Why showing PR but not ER? PT 10with 0,03%) and in methods the HER2 assessment.
Answers to Reviewer 2: We appreciate the positive feedback from the reviewer and the constructive suggestion for checking errors in Table 2. We have carefully control and noted that the values reported as a percentage score, mistakenly in the excel file have been transformed into numbers and therefore divided by 100. The value of 0.03 that Reviewer # 2 notes is, therefore, equal to 3, now in the revised table. The same check was done for the other values and changes were made in the table. We apologize for the lack of attention on an aspect that we recognize to be very important.ER positivity, for each sample, is reported in Figure 7B together with Ki-67 values, in Figure 7A.We revised section 4.9 in material and methods and introduced more details. The new text is highlighted in blue in the manuscript file:IHC primary antibodies used were: ERα (clone SP1, Ventana), PR (clone 1E2, Ventana), Ki67 (Clone 30-9, Ventana), Her2 (clone 4B5, Ventana). Tumors were considered positive for ER and PR when at least 1% of tumor cells showed unequivocal nuclear staining according to ASCO / CAP guidelines. PR expression was considered high in the presence of nuclear staining in 20% or more cells. We set a cut-off point to distinguish low versus high Ki67 expression at 20%. The original HER2/neu immunostained glass slides were concurrently reviewed by pathologists at a multiheaded microscope, and the consensus HER2/neu immunoreactivity was manually scored by conventional microscopy as 0, 1+, 2+, or 3+ according to the proposed HER /neu scoring system for breast cancer. According to the percentage of stained malignant cells, criteria for HER2 / neu score assignment were: 0, no staining or staining in <10% of cells; 1, faint staining in ≥10% of cells; 2, moderate staining in ≥10% of cells; and 3, strong staining in ≥10% of cells. Tumors classified as 0, 1+, and 2+ were considered “negative” and those scored as 3+ were classified as “positive”.